# Exceptional thermoelectric properties of flexible organic—inorganic hybrids with monodispersed and periodic nanophase

Liming Wang[1], Zimeng Zhang[1], Yuchen Liu[1], Biran Wang ◯ [1], Lei Fang[2], Jingjing Qiu[3], Kun Zhang ◯ [4] & Shiren Wang[1]

Flexible organic—inorganic hybrids are promising thermoelectric materials to recycle waste heat in versatile formats. However, current organic/inorganic hybrids suffer from inferior thermoelectric properties due to aggregate nanostructures. Here we demonstrate flexible organic—inorganic hybrids where size-tunable $Bi_2Te_3$ nanoparticles are discontinuously monodispersed in the continuous conductive polymer phase, completely distinct from traditional bi-continuous hybrids. Periodic nanofillers significantly scatter phonons while continuous conducting polymer phase provides favored electronic transport, resulting in ultrahigh power factor of ~1350 $\mu W\,m^{-1}\,K^{-2}$ and ultralow in-plane thermal conductivity of ~0.7 $W\,m^{-1}\,K^{-1}$. Consequently, figure-of-merit (ZT) of 0.58 is obtained at room temperature, outperforming all reported organic materials and organic—inorganic hybrids. Thermoelectric properties of as-fabricated hybrids show negligible change for bending 100 cycles, indicating superior mechanical flexibility. These findings provide significant scientific foundation for shaping flexible thermoelectric functionality via synergistic integration of organic and inorganic components.

[1] Department of Industrial and Systems Engineering, Texas A&M University, College Station, TX 77843, USA. [2] Department of Chemistry, Texas A&M University, College Station, TX 77843, USA. [3] Department of Mechanical Engineering, Texas Tech University, Lubbock, TX 77409, USA. [4] College of Textiles, Donghua University, 201620 Shanghai, China. Correspondence and requests for materials should be addressed to S.W. (email: s.wang@tamu.edu)

Currently, more than 60% of the primary energy of fossil fuels is lost worldwide as waste heat, and the loss is around 70% in automobiles[1]. This kind of energy loss brings a huge problem in energy utilization, and effectively recovering such waste heat is critical to overcome the energy crisis. Thermoelectrics involve conversion between heat and electricity, and thus it is extremely important for global sustainability. Integration of thermoelectric technology into industrial energy-utilization processes is of great interest and could provide an effective solution for power generation. Particularly, flexible materials are very attractive because of easy integration into various industrial processes. Flexible organic−inorganic hybrids, which can offer a huge degree of tunability because of diverse selections in both organic and inorganic components, have exhibited noteworthy promise for thermoelectric energy conversion[2–5]. Owing to the fascinating interfacial transport properties caused by the possible energy filtering effect and phonon scattering at the engineered nanoscale interfaces, organic−inorganic hybrids provide a new approach to avoid suffering from the trade-offs in thermoelectric parameters ($ZT = \sigma S^2 T/\kappa$, where $\sigma$ is electrical conductivity, $S$ is Seebeck coefficient, $T$ is absolute temperature, and $\kappa$ is thermal conductivity)[3,5–7]. In recent years, significant progress has been made on both $P$-type and $N$-type organic−inorganic hybrid thermoelectrics[3,5,8–13]. Highest ZT values of ~0.32 for $P$ type and ~0.2 for $N$ type (at room temperature) were achieved in poly(3,4-ethylenedioxythiophene) (PEDOT)/SnSe nanosheets composite films and organic molecules intercalated $TiS_2$ single crystals, respectively[5,9,14]. Furthermore, organic−inorganic hybrid thermoelectrics can be flexible and relatively light weight[1,8,15,16]. These features make them feasible to act as self-powered wearable devices by utilizing body's heat or other heat source to generate electricity, wearable temperature sensors, and flexible solid-state coolers, which are very difficult to achieve for inorganic thermoelectric materials since they are intrinsically brittle and rigid.

In principle, the thermoelectric parameters of organic−inorganic hybrids are highly dependent on the interfacial surface-to-volume ratio, as phonons and carriers are mainly scattered at interfaces in organic−inorganic hybrids. For example, enhanced thermoelectric power factor ($\sigma S^2$) was observed in nanowire-filled poly(3-hexylthiophene) (P3HT) rather than nanoparticle-filled P3HT as a result of the higher specific surface area of nanowires[12]. Nonetheless, the commonly encountered aggregation of inorganic nanoparticles in organic matrix led to a reduced interfacial area, and therefore hindered performance boosts, even making the thermoelectric properties of hybrids far below the calculated values based on series/parallel connected models[13]. As a result, the thermoelectric properties of present reported organic−inorganic hybrids are still much lower than the best commercial inorganic semiconductors[17]. It is highly desirable to develop organic−inorganic hybrids with rationally designed and controlled interfaces at the nano scale.

On the other hand, recent studies have demonstrated that highly conductive PEDOT films, the state-of-the-art organic thermoelectric materials, display high thermal conductivity larger than $1\,W\,m^{-1}\,K^{-1}$ [18,19]. Thus special attention is also required to reduce the thermal conductivity in order to achieve high ZT values. In this regard, fabricating organic−inorganic hybrids with unique nanostructure will be a good choice since nanostructure engineering can effectively decrease the thermal conductivity via phonon scattering[7,20,21]. Recently, ultralow thermal conductivity contributed by periodic nanostructures were reported in silicon nanomesh films and microporous metal-organic frameworks[22,23]. Inspired by these, it is probable to obtain high thermoelectric properties in organic−inorganic hybrids with monodispersed and periodic nanophase. Unfortunately, preparation of such

nanostructured hybrids is still technically challenging, and their thermoelectric properties are still unknown so far.

Here we presented a versatile method for fabricating flexible hybrids with monodispersed and periodic nanophase patterned by nanosphere lithography, in which both the size and spacing of nanophase were fine tuned from tens to hundreds of nanometers. PEDOT/$Bi_2Te_3$ hybrid was chosen as a model system for thermoelectric studies, not only due to the excellent properties of each component, but also because of the similar work functions of PEDOT and $Bi_2Te_3$ which could make high-energy carriers readily pass through the interfaces[16,24]. The prepared PEDOT/$Bi_2Te_3$ hybrid films displayed a large power factor by combining the high electrical conductivity of tosylate doped PEDOT and the high Seebeck coefficient of thermally evaporated $Bi_2Te_3$ as well as carrier-filtering effect at the nanoscaled PEDOT-$Bi_2Te_3$ interfaces. As a result of greatly reduced thermal conductivity owing to the interfacial phonon scattering, ZT value of the hybrids reached 0.58 at room temperature. Such nanostructured hybrids provide an excellent platform for exploring diverse high-performance organic−inorganic hybrid thermoelectrics.

## Results

**Fabrication and characterization of hybrid films.** In this paper, we used polystyrene (PS) nanosphere monolayer as a mask to pattern $Bi_2Te_3$ nanophase since nanosphere lithography has been demonstrated as an effective route for large-area periodic nanostructure arrays on both rigid and flexible substrates, with obvious advantages of low cost, high reproducibility, and good controllability on structural parameters[25–29]. The fabrication process of PEDOT/$Bi_2Te_3$ hybrid films is schematically illustrated in Fig. 1. The template-assisted nanofabrication was developed to produce monodispersed nanofillers in the continuous polymer host.

The corresponding structures were characterized by scanning electron microscopy (SEM) and are shown in Fig. 2. Starting with the fabrication of closely packed PS monolayer on substrate (Fig. 2a, b), reactive ion etching (RIE) was then adopted to decrease the diameter of PS nanospheres (Fig. 2c, d). After that, chrome (Cr) was coated by thermal evaporation followed by a lift-off process to form a nanomesh protected layer. The exposed $SiO_2$ was then removed by $CH_3/O_2$ plasma etching, receiving long-range periodic cylindric-like nanohole arrays, as shown in Fig. 2e. Then, $P$-type $Bi_2Te_3$ nanoparticle arrays were prepared by thermal evaporation method followed by dissolving $SiO_2$ template. Finally, PEDOT was polymerized with $Bi_2Te_3$ nanoparticles via a modified vapor-phase polymerization (VPP) process[30,31].

Obviously, the size and spacing of resultant $Bi_2Te_3$ nanoparticles were tunable, which were determined by the etching time and the diameter of pristine PS nanospheres. The size of PS nanospheres can be precisely controlled by adjusting the etching time as their diameters were gradually reduced with the etching time. In order to obtain PS nanosphere patterns with different diameters and spacings in a wide nanoscale range, PS nanospheres with three different diameters (100, 300, and 600 nm; see Supplementary Fig. 1) were etched for a series of times in this work. Taken as examples, Fig. 2f−j shows five different sizes of fabricated $Bi_2Te_3$ nanoparticle arrays by using 100 nm PS nanospheres as patterns. As-prepared PEDOT/$Bi_2Te_3$ hybrid films starting from 100, 300, and 600 nm PS nanospheres are denoted as PEDOT/$Bi_2Te_3$(100), PEDOT/$Bi_2Te_3$(300), and PEDOT/$Bi_2Te_3$(600) hybrid films, respectively. The surface SEM images of as-prepared hybrid films with different size and spacing of $Bi_2Te_3$ nanoparticles are exhibited in Fig. 2k−o for PEDOT/$Bi_2Te_3$(100) hybrid films and Supplementary Fig. 2 for PEDOT/

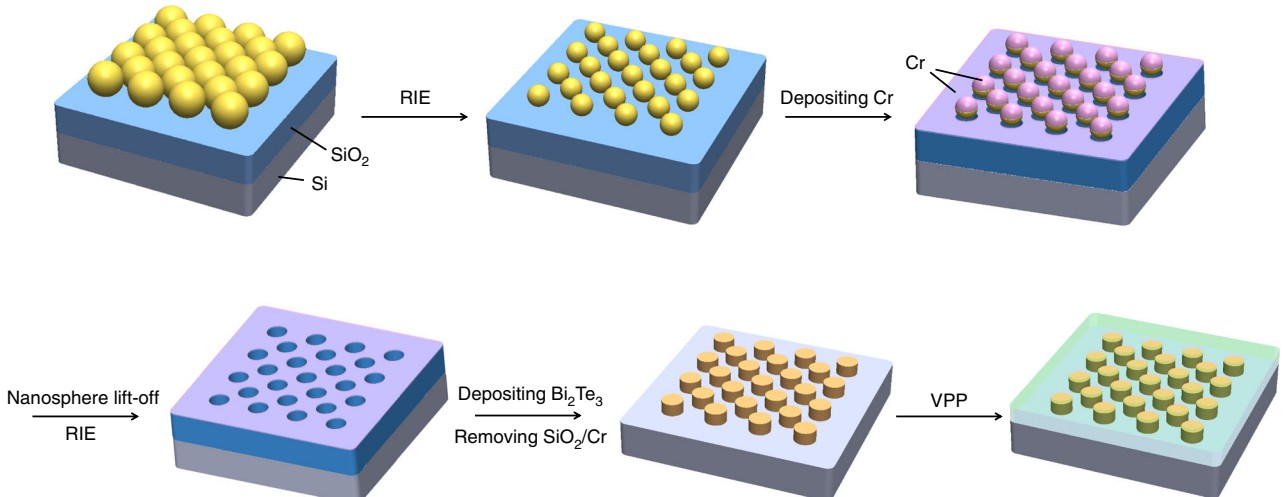

**Fig. 1** Scheme of controllable design for organic/inorganic hybrid film. Fabrication procedure for poly(3,4-ethylenedioxythiophene) (PEDOT)/Bi$_2$Te$_3$ hybrid films, including nanosphere lithography and reactive ion etching (RIE) for nanohole arrays template, filling Bi$_2$Te$_3$ into the template by thermal evaporation, removing the template, and compositing Bi$_2$Te$_3$ nanoparticle arrays with PEDOT by vapor-phase polymerization (VPP) process. The geometrical parameters of the resultant nanostructure can be tuned by the diameter of polystyrene (PS) nanospheres and the etching time for PS nanospheres

Bi$_2$Te$_3$(300) and PEDOT/Bi$_2$Te$_3$(600) hybrid films. Monodispersed and periodic Bi$_2$Te$_3$ nanophases in continuous PEDOT matrix were clearly observed in all PEDOT/Bi$_2$Te$_3$ hybrid films. The nanoparticle fraction was calculated according to the PS nanosphere packing model shown in Supplementary Fig. 3 (details are also provided in Supplementary Note 1), and digital photos of samples are shown in Supplementary Fig. 4. The X-ray diffraction (XRD) characterization shown in Supplementary Fig. 5 confirmed the Bi$_2$Te$_3$ crystal in the fabricated films.

**Thermoelectric properties**. The in-plane electrical conductivity, Seebeck coefficient, and power factor of PEDOT/Bi$_2$Te$_3$ hybrid films were characterized and the measurement schemes are shown in Supplementary Fig. 6 and Fig. 7 by in-house built systems. Calibration results of our systems are shown in Supplementary Fig. 8 and Fig. 9. The thermoelectric properties were investigated as a function of Bi$_2$Te$_3$ nanoparticle fraction, as shown in Fig. 3. Similar to literatures[30,31], VPP-fabricated PEDOT films displayed an extremely high electrical conductivity of ~1350 S cm$^{-1}$. With increasing Bi$_2$Te$_3$ nanoparticle fraction, the electrical conductivity of hybrid films greatly decreased while the Seebeck coefficient increased, since Bi$_2$Te$_3$ possessed a much higher Seebeck coefficient and lower electrical conductivity than PEDOT film (Supplementary Table 1). We calculated the electrical conductivity and Seebeck coefficient of hybrid films based on both parallel and series connected model[17,32],

$$\sigma(\text{parallel}) = \sigma_{\text{Bi}_2\text{Te}_3} x + \sigma_{\text{PEDOT}}(1 - x), \quad (1)$$

$$\sigma(\text{series}) = \frac{\sigma_{\text{Bi}_2\text{Te}_3} \sigma_{\text{PEDOT}}}{\sigma_{\text{Bi}_2\text{Te}_3}(1 - x) + \sigma_{\text{PEDOT}} x}, \quad (2)$$

$$S(\text{parallel}) = \frac{S_{\text{Bi}_2\text{Te}_3} \sigma_{\text{Bi}_2\text{Te}_3} x + S_{\text{PEDOT}} \sigma_{\text{PEDOT}}(1 - x)}{\sigma_{\text{Bi}_2\text{Te}_3} x + \sigma_{\text{PEDOT}}(1 - x)}, \quad (3)$$

$$S(\text{series}) = \frac{S_{\text{Bi}_2\text{Te}_3} \kappa_{\text{PEDOT}} x + S_{\text{PEDOT}} \kappa_{\text{Bi}_2\text{Te}_3}(1 - x)}{\kappa_{\text{PEDOT}} x + \kappa_{\text{Bi}_2\text{Te}_3}(1 - x)}, \quad (4)$$

where $\sigma(\text{parallel})$ and $S(\text{parallel})$ were the calculated electrical conductivity and Seebeck coefficient of the hybrids based on the

parallel connected model, $\sigma(\text{series})$ and $S(\text{series})$ were the calculated electrical conductivity and Seebeck coefficient of the hybrids based on the series connected model, and $x$ was the volume fraction of the Bi$_2$Te$_3$ nanoparticles in the hybrids. The electrical conductivity and Seebeck coefficient values of PEDOT and Bi$_2$Te$_3$ used in the models were based on the average values of more than three specimens. The results were illustrated in Fig. 3a, b (dash line). The electrical conductivity of all hybrid films stood within the calculated range, while the Seebeck coefficient of some PEDOT/Bi$_2$Te$_3$(100) hybrid films exceeded the calculated upper bound. In addition, the hybrid films also showed obviously size-dependent relationship in electrical conductivity and Seebeck coefficient. The highest Seebeck coefficient but the lowest electrical conductivity were observed in PEDOT/Bi$_2$Te$_3$(100) hybrid films.

All of these results indicated that the interfacial transport also played an important role in final thermoelectric properties of hybrids apart from the simple mixed effect. Energy filtering effect has been reported to improve the Seebeck coefficient in organic−inorganic hybrids in which interfaces form energy barriers that preferentially scattered low-energy carriers[7,16,33]. This can make the relaxation time strongly depend on energy and increase the asymmetry of carrier transport about the Fermi level, resulting in an enhanced Seebeck coefficient. The as-prepared hybrids containing the smallest nanoparticles exhibited the largest interfacial surface-to-volume ratio, providing the most sites to selectively scatter low-energy carriers due to the interfacial energy barrier between PEDOT and Bi$_2$Te$_3$, and thereby leading to the largest Seebeck coefficient. Additionally, saturated trend in the Seebeck coefficient was observed in PEDOT/Bi$_2$Te$_3$(100) hybrid films. Hence, a maximum power factor of ~1350 $\mu$W m$^{-1}$ K$^{-2}$ was achieved in PEDOT/Bi$_2$Te$_3$(100) hybrid films with ~31 vol% Bi$_2$Te$_3$ nanoparticles (Fig. 3c).

To understand the nanophase-dependent phonon transport, the in-plane thermal conductivity of fabricated films was measured by a differential $3\omega$ method[17,34,35], which was conducted in the same direction with the measured electrical conductivity and Seebeck coefficient. We fabricated thick hybrid films with a uniform thickness of ~0.75 $\mu$m (as shown in Supplementary Fig. 10) for the in-plane thermal conductivity measurement since it is challenging to perform in-plane thermal conductivity measurement for thin films less than 100 nm (see

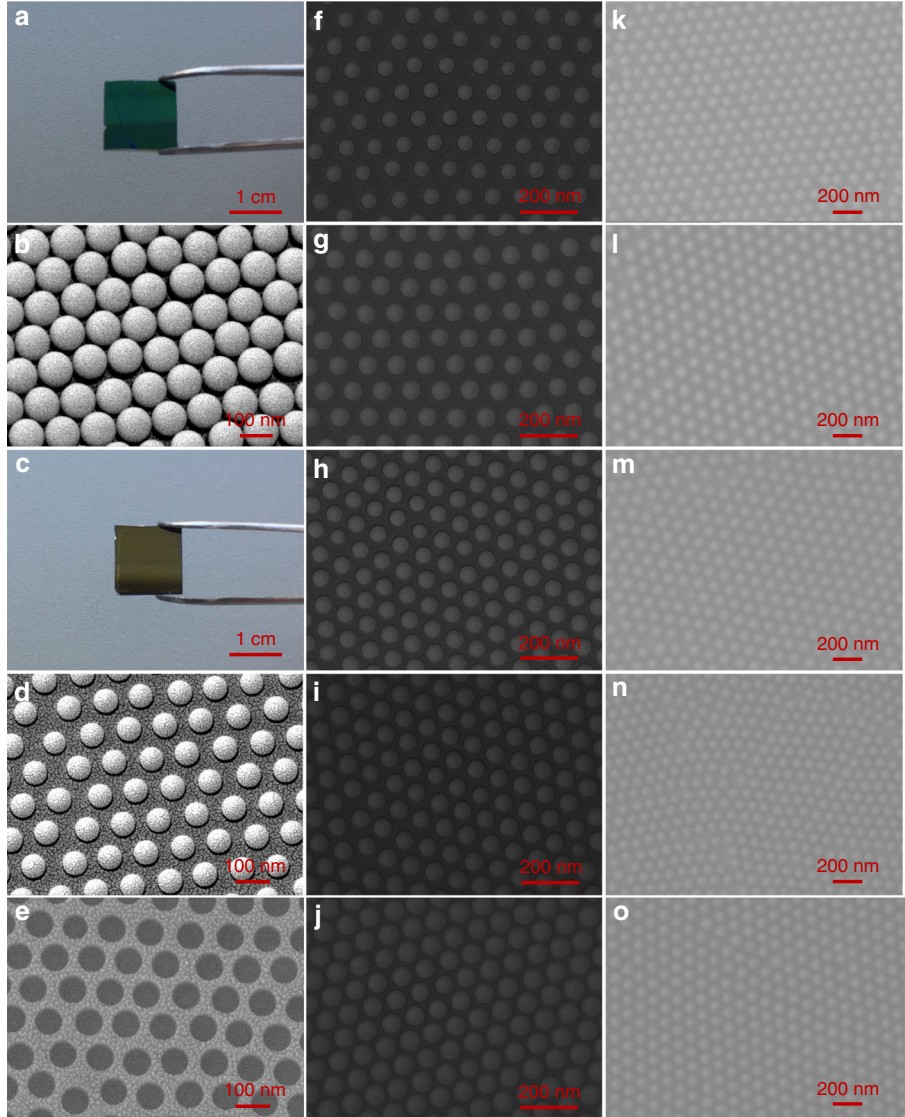

**Fig. 2** Morphological characterizations. **a** Digital photo of closely packed monolayer PS nanospheres with diameter of 100 nm on SiO$_2$/Si substrate. **b** Scanning electron microscopy (SEM) image of closely packed monolayer PS nanospheres with diameter of 100 nm. **c** Digital photo of PS nanospheres on SiO$_2$/Si substrate after etching for 22 s. **d** SEM image of PS nanospheres after etching for 22 s. **e** SEM image of prepared nanohole arrays template. **f**−**j** SEM images of Bi$_2$Te$_3$ nanoparticle arrays prepared by using 100 nm PS nanospheres as patterns. **k**−**o** SEM images of PEDOT/Bi$_2$Te$_3$(100) hybrid films

Supplementary Figs. 11, 12, and 13 and Note 4 for detailed preparation process and measurement)[34,35]. As-prepared neat PEDOT films (see Supplementary Note 4 in the Supplementary Information for detailed preparation process) showed a high in-plane thermal conductivity of 1.52 W m$^{-1}$ K$^{-1}$ at room temperature (Fig. 4a). After introduction of the monodispersed and periodic Bi$_2$Te$_3$ nanoparticles into PEDOT matrix, the thermal conductivity of PEDOT/Bi$_2$Te$_3$ hybrid film was greatly suppressed with Bi$_2$Te$_3$ nanoparticle fraction. The thermal conductivity for PEDOT/52 vol% Bi$_2$Te$_3$(100) hybrid film was only ~0.5 W m$^{-1}$ K$^{-1}$, about 300% reductions as compared to PEDOT film. More interestingly, this value was also lower than that of Bi$_2$Te$_3$ film (Supplementary Table 1), going against the mixture rule in classical composites where the thermal conductivity should be between that of the two components. In addition, the thermal conductivity of hybrid films was also related to the size of Bi$_2$Te$_3$ nanoparticles (Fig. 4b). With a similar volume fraction, hybrid films with smaller Bi$_2$Te$_3$ nanoparticles displayed a lower thermal conductivity. These results suggested that the great reduction in thermal conductivity for PEDOT/Bi$_2$Te$_3$ hybrid

films should be caused by not only the relatively lower thermal conductivity of Bi$_2$Te$_3$ fillers but also the interfacial effect. The unique nanostructure achieved in these hybrid films contributed to a larger interfacial surface-to-volume ratio, especially for the hybrids with smaller Bi$_2$Te$_3$ nanoparticles, and thereby leading to a lower thermal conductivity due to the interfacial effect.

Taking into account the interfacial effect as well as the size and volume fraction of nanoparticles, the effective thermal conductivity ($\kappa_{eff}$) of hybrids composed of continuous matrix and cylindric fillers (circular cylinders oriented perpendicularly to heat flow) can be estimated by a theoretical equation[36],

$$\kappa_{eff} = \kappa_m \frac{\left[\left(\frac{\kappa_d}{\kappa_m} - \frac{\kappa_d}{r_d h_c} - 1\right) V_d + \left(\frac{\kappa_d}{\kappa_m} + \frac{\kappa_d}{r_d h_c} + 1\right)\right]}{\left[\left(\frac{\kappa_d}{r_d h_c} - \frac{\kappa_d}{\kappa_m} + 1\right) V_d + \left(\frac{\kappa_d}{\kappa_m} + \frac{\kappa_d}{r_d h_c} + 1\right)\right]}, \quad (5)$$

where $\kappa_m$ is the thermal conductivity of the matrix (in this case PEDOT), $\kappa_d$ is the thermal conductivity of the dispersions (in this case Bi$_2$Te$_3$), $V_d$ is the volume fraction of dispersions, $r_d$ is the radius of circular cylindrical dispersions, and $h_c$ is the interfacial

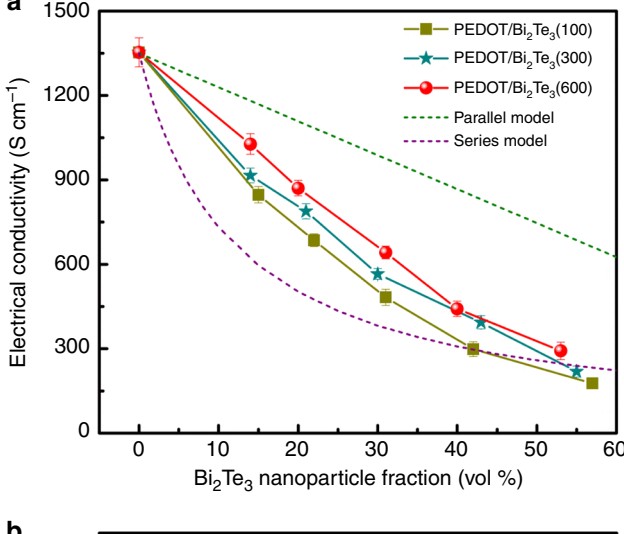

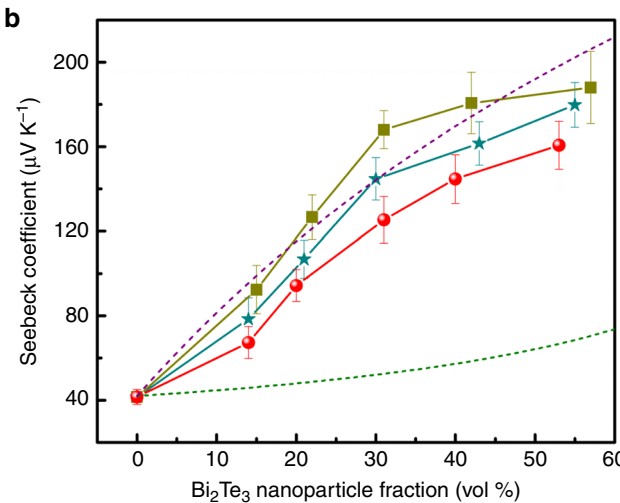

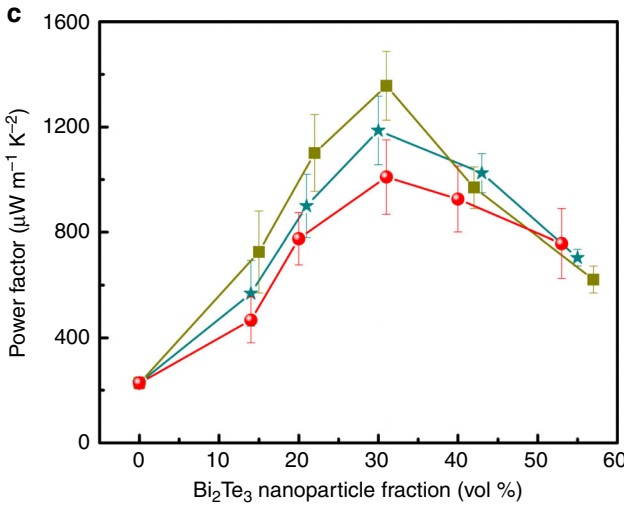

**Fig. 3** In-plane electrical conductivity, Seebeck coefficient, and power factor. **a** Electrical conductivity of PEDOT/Bi$_2$Te$_3$ hybrid films as a function of Bi$_2$Te$_3$ nanoparticle fraction. **b** Seebeck coefficient of PEDOT/Bi$_2$Te$_3$ hybrid films as a function of Bi$_2$Te$_3$ nanoparticle fraction. **c** Power factor of PEDOT/Bi$_2$Te$_3$ hybrid films as a function of Bi$_2$Te$_3$ nanoparticle fraction. Each point shows the standard deviations from three independent measurements

conductance. When $h_c = \infty$, this equation agrees with the expression of Maxwell for effective thermal conductivity of composites without considering the interfacial thermal resistance. The dash lines in Fig. 4b are theoretical values derived from Eq. (5) with a interfacial conductance of ~2.21 × 10$^5$ W m$^{-2}$ K$^{-1}$ (calculated with the experimental data of PEDOT/~31 vol% Bi$_2$Te$_3$(100) hybrid film). Our experimental data were well described by this model, since all the thermal conductivities of hybrid films were relatively consistent with theoretical ones. It is noted that the interfacial thermal conductance is very low which is at least two orders of magnitude lower than lots of reported values[37]. The low interfacial thermal conductance is possibly caused by the strong acoustic mismatch due to different phonon densities and velocities between polymer matrix (PEDOT) and inorganic filler (Bi$_2$Te$_3$)[3,38]. Furthermore, the phonon mean free path of PEDOT might be in 10~10$^2$ order of magnitude[39], which is comparable to the size of Bi$_2$Te$_3$ nanoparticles in the hybrid films. This will enhance the phonon scattering at interfaces and thereby the thermal transport can be suppressed[3,12]. Regarding the electrical transport, the matrix in hybrids is continuous and highly conductive PEDOT, which provides hole transport paths. More importantly, PEDOT is in situ polymerized with the presence of Bi$_2$Te$_3$ nanoparticles. This is good for intimate contact between PEDOT and Bi$_2$Te$_3$ and thereby enhancing charge transport across the PEDOT−Bi$_2$Te$_3$ interfaces[33]. Many works have reported in situ synthesized polymer composites with enhanced electrical transport properties[3,40]. Besides, the hole mean free path of PEDOT was reported to be more than one order of magnitude smaller than the phonon mean free path[39]. Thus, the hole transport is minimally affected as compared to the phonon transport. Precise experimental measurement of the interfaces at the nanoscale will be helpful to better illustrate the interfacial thermal/electrical conductance in the future work. As a result of large power factor and greatly reduced thermal conductivity, the PEDOT/Bi$_2$Te$_3$ hybrid film presented a maximum in-plane ZT value of ~0.58 at room temperature for PEDOT/31 vol% Bi$_2$Te$_3$(100) hybrid film (Fig. 4c), the highest ever shown for organic materials and organic/inorganic hybrids (Fig. 4d)[5,6,9,17,34,41–45]. The effect of Bi2Te3 particle sizes on the power factor and ZT is also shown in Supplementary Fig. 14. Air stability tests are shown in Supplementary Fig. 15.

**Mechanical flexibility**. The mechanical flexibility of PEDOT/Bi$_2$Te$_3$ hybrid film prepared on soft substrates was also investigated, and the photos of measurements are shown in Supplementary Fig. 16. When the PEDOT/31 vol% Bi$_2$Te$_3$(100) hybrid film was attached onto the surface of glass tube with different radius, the electrical resistance only slightly changed under bending deformation, within 5% of the initial value even at a very low curve radius of 3.5 mm (Fig. 5a). In contrast, directly deposited Bi$_2$Te$_3$ film by thermal evaporation displayed significant increases in electrical resistance with decrease of curve radius, reaching 18% change at curve radius of 3.5 mm. Furthermore, the hybrid film also showed higher mechanical stability compared with Bi$_2$Te$_3$ film, demonstrating negligible change in electrical resistance upon 100 bending cycles under curve radius of 3.5 mm, as illustrated in Fig. 5b. While the resistance of Bi$_2$Te$_3$ film continually increased with the bending times because of the appearance of micro-cracks in the Bi$_2$Te$_3$ film after bending (Supplementary Fig. 17). In the hybrid films, monodispersed and periodic Bi$_2$Te$_3$ nanoparticles were homogenously surrounded by flexible PEDOT. The intimate contact between Bi$_2$Te$_3$ nanoparticles and PEDOT matrix can accommodate deformations and thereby contributes to the superior flexibility. This also can be seen in Supplementary Fig. 18, the hybrid film with the largest

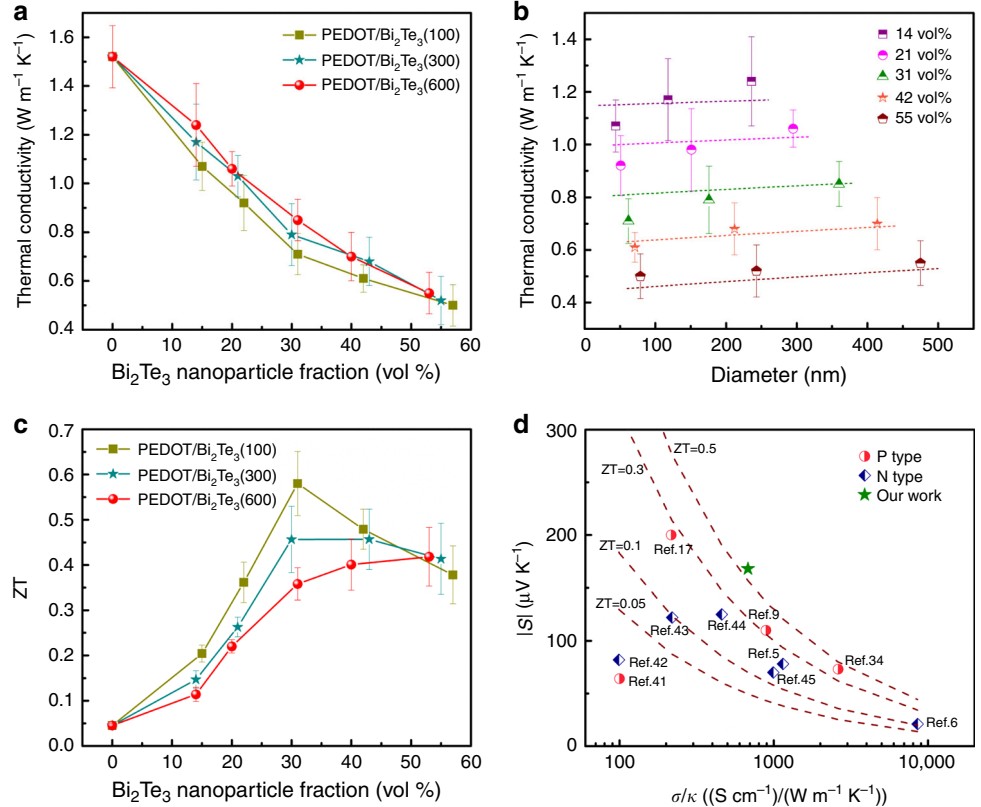

**Fig. 4** In-plane thermal conductivity and ZT value. **a** Thermal conductivity of PEDOT/Bi$_2$Te$_3$ hybrid films as a function of Bi$_2$Te$_3$ nanoparticle fraction. Each point shows the standard deviations from three independent measurements. **b** Thermal conductivity of PEDOT/Bi$_2$Te$_3$ hybrid films as a function of the diameter of Bi$_2$Te$_3$ nanoparticles. The dash lines are the fitting lines based on Eq. (5) under a given Bi$_2$Te$_3$ nanoparticle fraction. **c** ZT value of PEDOT/Bi$_2$Te$_3$ hybrid films as a function of Bi$_2$Te$_3$ nanoparticle fraction. **d** Comparison of the thermoelectric properties around room temperature between this work and the state-of-the-art organic materials and organic–inorganic composite materials, including conducting polymers[17, 34], conducting polymer-inorganic composites[9], chemically doped single-walled carbon nanotubes[41, 42], metal coordination polymers[43, 44], organic intercalated transition metal dichalcogenides[5], insulating polymer-metal composites[6], and organic–inorganic hybrid perovskites[45]

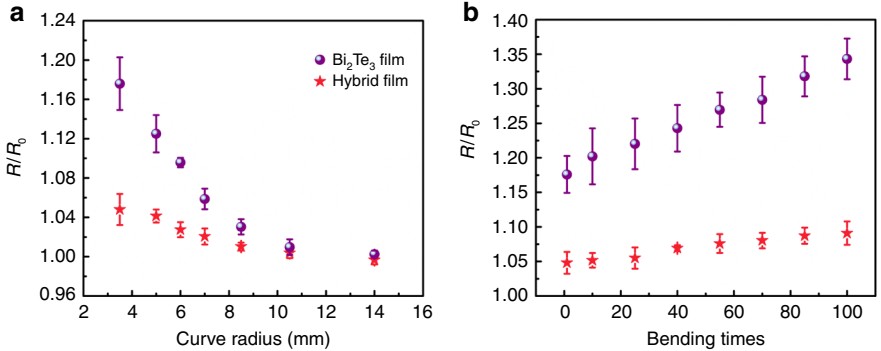

**Fig. 5** Mechanical flexibility. Comparison of the flexibility between PEDOT/Bi$_2$Te$_3$(100) hybrid films with ~31 vol% Bi$_2$Te$_3$ nanoparticle fraction (red pentacle) and Bi$_2$Te$_3$ films (purple circle). **a** The resistance $R$ of the film as a function of curve radius $r$, where $R_0$ is the resistance before bending. **b** The resistance of the films as a function of bending times with a curve radius of 3.5 mm. Each point shows the standard deviations from three independent measurements. All films were prepared on flexible polyimide substrates

interfacial surface-to-volume ratio showed the best mechanical flexibility. In addition to the bending effect on the electrical conductivity, the bending radius effect on the Seebeck coefficient was also examined as shown in Supplementary Fig. 19. These results demonstrate the flexible hybrid thermoelectric films are possible to be tailored as new types of devices outside of the domain of conventional rigid inorganic thermoelectric materials.

## Discussion

In summary, flexible PEDOT/Bi$_2$Te$_3$ hybrid films with monodispersed and periodic Bi$_2$Te$_3$ nanophase have been successfully fabricated, exhibiting an ultrahigh ZT value of ~0.58 at room temperature. The unique nanostructure in hybrid films supplies an optimized interfacial surface-to-volume ratio, which not only contributes to a greatly reduced thermal conductivity but also the good mechanical flexibility. These results indicate the importance

of nanostructure engineering when exploring hybrid thermo-electric materials with both high performance and mechanical flexibility. Owing to the versatility of the fabrication method, numerous organic—inorganic hybrids also can be obtained by replacing the organic or inorganic component, providing a plat-form for creating nanostructured hybrid materials with wide applications.

## Methods

**Materials**. Monodispersed suspension of PS nanospheres (10 wt%, in water) with various diameters (100, 300, and 600 nm), sodium dodecyl sulfate (SDS, ≥99%), 3,4-ethylenedioxythiophene (EDOT, 97%), poly(ethylene glycol)-block-poly(pro-pylene glycol)-block-poly(ethylene glycol) (PEG-PPG-PEG, $M_w = 5800$), dime-thylformamide (DMF), $n$-butanol and toluene were purchased from Sigma-Aldrich. Iron (III) tosylate (Fe(Tos)$_3$, 40 wt%, in $n$-butanol) (Clevios$^{TM}$ C-B 40 V2) were purchased from H.C. Starck. P-type Bi$_2$Te$_3$ powders (99.99%, ~1250 mesh) were purchased from KYD Materials. Silicon wafers (undoped, resistivity $\rho$ > 10,000 ohm cm) were purchased from University Wafer. Flexible polyimide sub-strates were purchased from Gizmodorks. All the materials were used as received.

**Pretreatment of substrates**. Si wafers were cut into pieces (13 mm × 13 mm) and deposited with SiO$_2$ layer (100 nm in thickness) by plasma-enhanced chemical vapor deposition (Oxford Plasmalab 80 Plus). The substrates were successively immersed into acetone, ethanol, and DI water by sonication, each for 15 min. Then the substrates were sonicated in piranha solution for 30 min to make the surface hydrophilic. After being washed with DI water, the substrates were dried under nitrogen.

**Fabrication of monolayer PS nanosphere**. One clean substrate was placed at the mid-bottom and several clean substrates were placed at the edges of a Petri dish with a diameter of 10 cm. The height of the middle substrate was higher than the height of the substrates at the edges. Then DI water was added carefully to the dish around the middle substrate, until the water covered the edge of the upper surface of the substrate. Monodispersed suspension of PS nanospheres was diluted with an equal volume of ethanol. Then 50 μL of the dispersion was dropped on the top of the substrate, which spread freely to cover nearly the whole water surface within a few seconds. Twenty microliters of SDS solution (1 wt%) was dropped onto the water surface to reduce the surface tension and make the PS spheres pack closer. Then the dish was slowly sloped at 10 degrees. The monolayer PS spheres were transferred onto substrates after extracting the water in the dish. Finally the sub-strates were dried in air at room temperature.

**Fabrication of Bi$_2$Te$_3$ nanoparticle arrays**. Firstly, RIE (Oxford Plasmalab 100 Plus) was used to reduce the diameter of the PS nanospheres. The etching was conducted under temperature of 20 °C, a chamber pressure of 20 mTorr, O$_2$/CHF$_3$ gas mixture (gas flow rates of 100/4.3 sccm), and a radio frequency (RF) power of 200 W. The etching time were 10, 16, 22, 27, and 32 s for 100 nm PS nanospheres, 34, 43, 52, 60, and 70 s for 300 nm PS nanospheres, 55, 66, 75, 85, and 96 s for 600 nm PS nanospheres, respectively. Then, 10-nm-thick Cr layer was thermally evaporated on the surface of monolayer PS nanospheres. The PS nanospheres were lifted off by ultrasonicating the substrates in toluene for 5 min at 35 °C. After drying, the samples were etched to remove the exposed SiO$_2$ layer by RIE under temperature of 35 °C, a chamber pressure of 45 mTorr, O$_2$/CHF$_3$ gas mixture (gas flow rates of 5/45 sccm), a radio frequency (RF) power of 150 W, and an etching time of 3 min. Subsequently, 70-nm-thick Bi$_2$Te$_3$ films were coated onto the as-prepared substrates by thermal evaporation under a chamber pressure of $10^{-5}$ mbar and deposition rate of 0.3 nm/s. Then the coated Bi$_2$Te$_3$ films were annealed at 250 °C for 1 h under vacuum. Finally, substrates with Bi$_2$Te$_3$ nano-particle arrays were obtained by removing SiO$_2$ layer with 4% HF solution.

**Preparation of PEDOT/Bi$_2$Te$_3$ hybrid films**. PEDOT/Bi$_2$Te$_3$ hybrid films were prepared by VPP process. Two grams Fe(Tos)$_3$ solution (40 wt%, in $n$-butanol), 1.5 g PEG-PPG-PEG, 1.5 g DMF, and 1.5 g $n$-butanol were thoroughly mixed by ultrasonication. The mixture was placed on a hotplate at 35 °C for 2 min prior to pipetting solution. Subsequently, the mixture was dropped and the entire substrate was covered with Bi$_2$Te$_3$ nanodisc arrays. The substrate was spin-coated at 4500 rpm for 45 s and immediately transferred onto a hotplate at 70 °C for 30 s. The substrate was transferred to a vacuum oven with a crucible containing EDOT monomers at the bottom of the chamber. The VPP process was conducted in the chamber at 35 °C under a pressure of 45 mbar for 25 min. After polymerization, the hybrid film was placed on a hotplate at 70 °C for 2 min and then cooled down to room temperature. The resultant thin film was thoroughly washed with ethanol to remove any oxidant residuals. The washed film was finally dried in air for further characterizations. Neat PEDOT thin films with thickness of ~56 nm were also prepared with the same conditions.

**Preparation of flexible PEDOT/Bi$_2$Te$_3$ hybrid films**. Flexible hybrid films were prepared following the above procedures by using flexible polyimide substrates.

**Characterizations**. The obtained films were characterized by field emission scanning electron microscope (JEOL JSM-7500F), XRD (Bruker D8) with a Cu-K$_a$ source, and optical microscopy (Olympus). The thermoelectric properties of pre-pared films were tested with home-built apparatus. Details of the measurements for in-plane electrical conductivity, Seebeck coefficient, and thermal conductivity were mentioned in Supplementary Notes 2–4. The mechanical flexibility was assessed by attaching the films on glass tubes with different diameter and testing their resistances.

## Data availability

The data that support the findings of this study are available from the corresponding author on request.

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

## Acknowledgements

S.W. appreciates the support from startup funds of Texas A&M University and TEES, Water initiative grant, and the National Science Foundation grant (CMMI 1634858). L.F. also acknowledges the fund support form Qatar National Priority Research Program. K.Z. acknowledges the financial supports from the National Natural Science Foundation of China (No.51603036), and Young Elite Scientists Sponsorship Program by CAST (2017QNRC001).

## Author contributions

S.W. conceived the research; L.W. and S.W. designed the experiments. L.W., Y.L., and B.W. prepared samples and measured the thermoelectric properties. L.W. and Z.Z. performed the other characterizations. L.W., L.F., J.Q., K.Z., and S.W. analyzed the data and wrote the manuscript with comments and inputs from all authors.

## Additional information

**Competing interests:** The authors declare no competing interests.

