## [Peer Review File · Nature Communications]

Reviewers' comments:

Reviewer #1 (Remarks to the Author):

Organic/inorganic thermoelectric hybrids are promising candidates as flexible power devices for future electronics to harvest waste heat. Current organic/inorganic hybrids suffer from inferior TE performance due to aggregate nanostructures. In this article, the authors fabricate flexible organic/inorganic hybrid films with monodispersed, size-tunable and periodic Bi₂Te₃ nanoparticles embedded in the conductive polymer matrix, which exhibit a high power factor and reduced thermal conductivity. A zT of 0.58 at room temperature has been achieved, much higher than other existing flexible materials. They also demonstrate the superior mechanical flexibility of the films. In my opinion, this work is novel, well-organized and presented. It should have a broad interest to the audience of this journal. I recommend to publish it in Nat Commun after minor revisions or some clarifications.

1. Thermoelectric properties of PEDOT film sometimes change from sample to sample, depending on processing. What values of PEDOT films are used in the mixture model for the calculation of Seebeck and electrical conductivity of hybrid films?

2. The uniform thick films were used for in-plane thermal conductivity measurement by a 3w method since it is a challenging to perform the measurement of films with a thickness less than 100nm. In this case, please clarify if the electrical conductivity and Seebeck coefficient were also measured on the same thick films?

3. The air stability test was lasted up to only 24h. From Figure S12, the electrical conductivity decays slightly. How about the TE performance if extending the test up to one week?

Reviewer #2 (Remarks to the Author):

In this manuscript, Wang et al reported flexible organic/inorganic hybrids with monodispersed and periodic nanophase. In fact, an organic/inorganic hybrid thermoelectric material is an attractive research area and the development of effective strategy to improve the TE performance is of vital importance. As for the hybrid materials, the conventional introduction of nanoparticles usually suffers from aggregation problem, leading to a challenge in fine-tuning the interface properties. In this work, the authors fabricated the hybrid films with PEDOT and fine-tuned Bi₂Te₃ nanoparticles. An exciting result claimed is the achievement of a high power factor over 1000 $\mu\text{W m}^{-1} \text{K}^{-2}$ and low thermal conductivity of 0.7 $\text{W m}^{-1} \text{K}^{-1}$. The concept and the experimental results are interesting, but I have the following concerns about the main claim of the authors.

1: The high performance is the main claim of this manuscript. Notably, the VPP deposited PEDOT (without Bi₂Te₃) show a high conductivity of 1350 S cm^{-1} and high Seebeck coefficient of 40 $\mu\text{V/K}$, which yield a high power factor over 200 $\mu\text{W m}^{-1} \text{K}^{-2}$ (Fig. 3). This performance is very high compared with previous reports without fine-tuned doping level (Nat. Mater. 2011, 10, 429). How could this happen? A discussion should be added.

2: As mentioned above, one feature in this study is that the Seebeck coefficient is large. So, the authors should make extra effort to motivate that the measurement of Seebeck coefficient is free of major errors. There are errors from the temperature measurement and from the voltage reading due to geometrical parameters (not due to instrumentation). The authors should show estimate the error on the Seebeck measurement due to the geometrical feature of the electrodes chosen according to the article [S. van Reenen, M. Kemerink, Organic Electronics, 15 (2014) 2250]. The device geometry for the measurement of Seebeck coefficient should be provided. Moreover, I suggest the deposition of thermal resistor on the hybrid film to measure the temperature difference. The utilization of thermocouple can easily lead to contact problem and result in measurement errors in temperature difference.

3: As for the measurement of thermal conductivity, it seems that it is not so easy to fabricate

thick film (over 500 nm) with method illustrated by Fig. 1. The authors should provide detailed fabricated method, whereas the characterization details are also needed to confirm that the thick film have a same or similar morphology or structure. What is Seebeck coefficient and electrical conductivity of the thick sample? By the way, what is thickness of PEDOT film in a typical thin film and so-called thick film.

4 : In the section of "Estimation of the Bi₂Te₃ nanoparticle fraction in hybrid films" in supplementary information, the thickness of hybrid films and Bi₂Te₃ nanoparticle are presented by d₀ and d₁, respectively. What is the relationship among d₀, d₁ and the 'typical average thickness of film' ('Electrical conductivity measurement' section in supplementary information). How to determine the exact thickness of the sample if these three thickness are different?

5: The SEM image of the device for thermal conductivity measurement should be provided in Fig s14.

6: As for the mechanical flexibility, I recommend the authors to perform the measurement of Seebeck coefficient upon different bending to make the result more convincing.

7: The fourth step in Figure 1 schematically showed that the SiO₂ is removed and Bi₂Te₃ is directly contact with Si substrate. I'm wondering whether the TE material is directly attached to Si substrate or not. Would the conductive Si substrate influence the Seebeck coefficient or electrical conductivity characterization? It is necessary to fabricate material on insulating substrate like glass to measure the TE performance.

Reviewer #3 (Remarks to the Author):

The presented hybrid film for thermoelectricity is interesting, and the fabrication method is also interesting. However, I have the following comments.

* Flexible materials with high ZT

The authors mentioned that flexible materials with high ZT doesn't exist in the abstract. However, the flexible Bi₂Te₃ films in both p and n type had been presented as follows;

K. Kato et al., Journal of electronic materials, Vol.43, No.6 (2014) pp.1733-1739.

* For N-type

The present study is only for p-type. N-type material is favorable to make a thermoelectric module.

* Interfacial resistance

The electrical conductivity of the composite follow the highest predicted value in the present study, however, those values follow the lowest value mostly mentioned below (sometime, lower than the lowest value due to interfacial electrical resistance).

B. Zhang et al., ACS Applied Materials & Interfaces, Vol.2, No.11 (2010) pp.3170-3178.

On the other hand, interfacial thermal resistance is very high. The order is 10⁻⁵ K/W/m².

Thermal resistance is well reviewed in the following article. The order of the resistance is 10⁻¹⁰ to 10⁻⁷ K/W/m².

T. Zhan et al., Scientific report, Vol.7 (2017) 7109.

The electrical resistance is really low although the thermal resistance is 100 times higher than other reported value at the present study. The mechanism of the present study should be explained to have the readers' understanding.

Response to Referees

Reviewer #1:

Comment 0: *In my opinion, this work is novel, well-organized and presented. It should have a broad interest to the audience of this journal. I recommend to publish it in Nat Commun after minor revisions or some clarifications.*

Response: Thank you for your positive comments and recommendation!

Comment 1: *Thermoelectric properties of PEDOT film sometimes change from sample to sample, depending on processing. What values of PEDOT films are used in the mixture model for the calculation of Seebeck and electrical conductivity of hybrid films?*

Response: The electrical conductivity and Seebeck coefficient values of PEDOT films used in the mixture model are the average values of more than three specimens. The values and error bars are shown in Fig. 3a and 3b. The VPP process of preparation of PEDOT for all PEDOT films as well as PEDOT/Bi₂Te₃ hybrid films were conducted under the same conditions. We added the sentence “The electrical conductivity and Seebeck coefficient values of PEDOT and Bi₂Te₃ used in the models were based on the average values of more than three specimens” on Page 7 in the revised manuscript.

Comment 2: *The uniform thick films were used for in-plane thermal conductivity measurement by a 3 ω method since it is a challenging to perform the measurement of films with a thickness less than 100 nm. In this case, please clarify if the electrical conductivity and Seebeck coefficient were also measured on the same thick films?*

Response: The in-plane electrical conductivity and Seebeck coefficient were measured on the thin films with thickness less than 100 nm. Since it is very difficult to perform the in-plane thermal conductivity measurement of films with a thickness less than 100 nm, thus we prepared thick hybrid films (thickness of ~ 0.75 μm) for thermal conductivity measurement by a 3 ω method, in order to estimate the ZT values of as-prepared hybrid films. All of these have been clarified in the section of “thermal conductivity measurement” in the revised Supplementary Information.

Comment 3: The air stability test was lasted up to only 24h. From Figure S12, the electrical conductivity decays slightly. How about the TE performance if extending the text up to one week?

Response: The air stability has been tested by extending the duration up to one week according to your suggestion. As shown in the following figure, all the films were relatively stable in air at room temperature, displaying slightly changes in electrical conductivity after one week. These results were added in Supplementary Fig. 15 in the revised Supplementary Information.

Figure | Air stability test of prepared PEDOT/Bi₂Te₃(100) hybrid film (square), PEDOT/Bi₂Te₃(300) hybrid film (pentacle), and PEDOT/Bi₂Te₃(600) hybrid film (circle) with ~31 vol% Bi₂Te₃.

Reviewer #2:

Comment 0: An exciting result claimed is the achievement of a high power factor over $1000 \mu W m^{-1} k^{-2}$ and low thermal conductivity of $0.7 W m^{-1} K^{-1}$. The concept and the experimental results are interesting.

Response: Thank you very much for the positive comments!

Comment 1: The high performance is the main claim of this manuscript. Notably, the VPP deposited PEDOT (without Bi₂Te₃) show a high conductivity of $1350 S cm^{-1}$ and high Seebeck coefficient of $40 \mu V/K$, which yield a high power factor over $200 \mu W m^{-1} k^{-2}$ (Fig. 3). This performance is very high compared with previous reports without fine-tuned doping level (Nat. Mater. 2011, 10, 429). How could this happen? A discussion should be added.

Response: The VPP process used in this work is referred to another paper (Nat. Mater. 2014, 13, 190), as we mentioned on Page 5 in the revised manuscript that “PEDOT was polymerized with

Bi₂Te₃ nanoparticles via a modified vapor-phase polymerization (VPP) process^{30,31}.” In the reference, the electrical conductivity of VPP deposited PEDOT films can be 1200~1500 S cm⁻¹, and the Seebeck coefficient is in the range of 35~50 μV K⁻¹. Our results are comparable to previously reported values in the references.

Comment 2: *As mentioned above, one feature in this study is that the Seebeck coefficient is large. So, the authors should make extra effort to motivate that the measurement of Seebeck coefficient is free of major errors. There are errors from the temperature measurement and from the voltage reading due to geometrical parameters (no due to instrumentation). The authors should show estimate the error on the Seebeck measurement due to the geometrical feature of the electrodes chosen according to the article [S. van Reenen, M. Kemerink, Organic Electronics, 15 (2014) 2250]. The device geometry for the measurement of Seebeck coefficient should be provided. Moreover, I suggest the deposition of thermal resistor on the hybrid film to measure the temperature difference. The utilization of thermocouple can easily lead to contact problem and result in measurement errors in temperature difference.*

Response: Thank you for providing us with this important article (Organic Electronics, 2014, 15, 2250). We strongly agree with you that the temperature measurement and the voltage reading due to geometrical parameters will greatly affect the accuracy of the Seebeck coefficient measurement. In fact, we also carefully read this paper before and thereby chose the geometrical feature of the electrodes according to this article. Similar to this article as well as a previous paper reported by our group (Nanoscale, 2016, 8, 8033. Supporting Information), a set of parallel and narrow line-shaped (1 mm × 7 mm) gold electrodes with thickness of 150 nm and spacing of 10 mm were used, in order to get an accurate determination of the actual Seebeck coefficient. In addition, both the thermocouples and the sample surface in the region of the thermocouples were erased by a swab with ethanol each time in order to avoid the contact problem and get accurate measurement in temperature difference. The details for the measurement of Seebeck coefficient have been added in the section “Seebeck coefficient measurement” in the revised Supplementary Information. More importantly, two different reference samples, n-type Bi₂Te₃ sample with Seebeck coefficient of -180 μV K⁻¹ and constantan with Seebeck coefficient of -36 μV K⁻¹ which were obtained by a well-calibrated commercial instrument ZEM-3, were tested in order to confirm the accuracy of this method. The Seebeck coefficient measurement curves of this two

samples are illustrated as follows. It reveals the measurement errors for Seebeck coefficient were less than 6%.

Figure | The measured Seebeck coefficient of standard samples: (a) n-type Bi₂Te₃ and (b) constantan.

Comment 3: *As for the measurement of thermal conductivity, it seems that it is not so easy to fabricated thick film (over 500 nm) with method illustrated by Fig. 1. The authors should provide detailed fabricated method, whereas the characterization details are also needed to confirm that the thick film have a same or similar morphology or structure. What is Seebeck coefficient and electrical conductivity of the thick sample? By the way, what is thickness of PEDOT film in a typical thin film and so-called thick film.*

Response: Although the electrical conductivity and Seebeck coefficient were measured from thin films with thickness less than 100 nm, it is still very difficult to perform the in-plane thermal conductivity measurement for these thin films. Thus we prepared thick hybrid films (thickness of ~0.75 μm) for thermal conductivity measurement by a 3ω method. Compared with the method for preparing thin hybrid films, several modifications were made to prepare thick hybrid films. Firstly, 1 μm thick SiO₂ layer was deposited on the Si wafer via PEVCD in the step of “pre-treatment of substrates”. Then, the etching time for removing the exposed SiO₂ by RIE was increased to 30 min in the step of “fabrication of Bi₂Te₃ nanoparticle arrays”. Also, the thickness of deposited Bi₂Te₃ film was fixed to 0.7 μm. Finally, the parameters of spin coating in the step of VPP was changed to 1500 rpm for 25 s. The VPP process was repeated for three times to obtain desired thick hybrid films. Taken as an example, the SEM image of as-prepared PEDOT/31 vol% Bi₂Te₃(100) hybrid thick film is given in Supplementary Fig. 10 in the revised

Supplementary Information, which displays a very similar morphology as compared to the corresponding thin film. All of the above discussions have been provided in the section of “thermal conductivity measurement” in the revised Supplementary Information according to your comment. Both the electrical conductivity and Seebeck coefficient of the as-prepared thick films are just a little lower than the corresponding thin film. For instance, the electrical conductivity and Seebeck coefficient of PEDOT/31 vol% Bi₂Te₃(100) hybrid thick film are 391 S cm⁻¹ and 152 μV K⁻¹. The thickness of as-prepared PEDOT thin film is ~56 nm. We modified the Method section with proper revision in the revised manuscript (Page 15). For the PEDOT thick films for thermal conductivity measurement, the detailed preparation method was also provided in the section of “thermal conductivity measurement” in the revised Supplementary Information. The thickness of as-prepared PEDOT thick film was measured to be ~0.62 μm.

Comment 4: *In the section of “Estimation of the Bi₂Te₃ nanoparticle fraction in hybrid films” in supplementary information, the thickness of hybrid films and Bi₂Te₃ nanoparticle are presented by d_0 and d_1 , respectively. What is the relationship among d_0 , d_1 and the ‘typical average thickness of film’ (‘Electrical conductivity measurement’ section in supplementary information). How to determine the exact thickness of the sample if these three thickness are different?*

Response: We apologize for the misleading descriptions in our manuscript. The thickness of Bi₂Te₃ nanoparticle (d_1) can be accurately controlled by the monitor of thermal evaporation instrument, which is also confirmed by a Bruker DektakXT surface profiler. The thickness of Bi₂Te₃ nanoparticle (d_1) is ~70 nm as mentioned in the section of “Fabrication of Bi₂Te₃ nanoparticle arrays” on Page 14 in the manuscript. The thickness of hybrid films (d_0) is mainly determined by the spin coating conditions of VPP process. For hybrid films, we measured the film thickness in at least five different areas for each sample with a Bruker DektakXT surface profiler. The thickness of hybrid films (d_0) is ~75 nm.

In the section of “Electrical conductivity measurement” in Supplementary Information, the typical average thickness of film means the thickness of hybrid films (d_0). Indeed, these thicknesses show a little difference. Thus, in our work, we estimated the Bi₂Te₃ nanoparticle fraction in hybrid films and calculated the electrical conductivity of all films with the tested film thickness.

Thank you for your valuable comment. We have made revisions in the section of “Estimation of the Bi_2Te_3 nanoparticle fraction in hybrid films” and deleted the misleading sentence “typical average thickness of film” in the section of “Electrical conductivity measurement” in the revised Supplementary Information.

Comment 5: *The SEM image of the device for thermal conductivity measurement should be provided in Fig s14.*

Response: Thank you for your suggestion. The characterizations of the device for thermal conductivity measurement (see below) have been provided in the revised Supplementary Information (Supplementary Fig. 12).

Figure 1 | (a, b) Cross-sectional SEM images of device for the thermal conductivity measurement, (a) sample and (b) reference. (c, d) Optical images of gold line with width of (c) 1 μm and (d) 20 μm . The scale bars in a, b, c and d are 1 μm , 1 μm , 2 μm and 10 μm , respectively.

Comment 6: *As for the mechanical flexibility, I recommend the authors to perform the measurement of Seebeck coefficient upon different bending to make the result more convincing.*

Response: According to your suggestion, the Seebeck coefficient of PEDOT/31 vol% $\text{Bi}_2\text{Te}_3(100)$ hybrid film upon different bending radii were measured. The results were listed as following. The temperature difference was generated by heating one side of the hybrid film with a flexible heater which was connected with a Keithley 2400 SourceMeter. The Seebeck coefficient kept stable upon different bending radii, only ~10% changes even at a very low curve radius of 3.5 mm. These results were added in the revised Supplementary Information (Supplementary Fig. 19).

Figure | The Seebeck coefficient S of prepared PEDOT/Bi₂Te₃(100) hybrid film with ~31 vol% Bi₂Te₃ nanoparticle fraction as a function of curve radius, where S_0 is the Seebeck coefficient before bending.

Comment 7: *The fourth step in Figure 1 schematically showed that the SiO₂ is removed and Bi₂Te₃ is directly contact with Si substrate. I'm wondering whether the TE material is directly attached to Si substrate or not. Would the conductive Si substrate influence the Seebeck coefficient or electrical conductivity characterization? It is necessary to fabricate material on insulating substrate like glass to measure the TE performance.*

Response: The etching condition for SiO₂ layer by RIE was negligibly affected the Si wafer. After completely removing exposed SiO₂, Bi₂Te₃ was directly deposited on exposed Si substrate. Then, SiO₂ (under Cr protective layer) can be easily and fastly dissolved by HF solution in the fourth step, leaving Bi₂Te₃ nanoparticle arrays on the Si substrates which was confirmed by SEM images (Fig. 2f-j).

In order to exclude the effect of substrate, we used nonconductive Si wafer as substrate, as mentioned in the Method. We are sorry for the unclear expression. The detailed information of used Si wafer is provided on Page 13 in the revised manuscript: Silicon wafers (undoped, resistivity $\rho > 10000$ ohm cm) were purchased from University Wafer. Before using the silicon wafers, we also tested and confirmed that these silicon wafers were nonconductive.

Reviewer #3:

Comment 0: *The presented hybrid film for thermoelectricity is interesting, and the fabrication method is also interesting.*

Response: Thank you very much for your positive comments.

Comment 1: *Flexible materials with high ZT. The authors mentioned that flexible materials with high ZT doesn't exist in the abstract. However, the flexible Bi₂Te₃ films in both p and n type had been presented as follows; K. Kato et al., Journal of electronic materials, Vol.43, No.6 (2014) pp.1733-1739.*

Response: Thank you for your comment. To make accurate statement, we modified the sentence from “such a ZT value is much higher than any existing flexible materials” to “outperforming all reported organic materials and organic/inorganic hybrids” in the Abstract. In our work, we report flexible organic/inorganic hybrids with high thermoelectric properties, which hold lower density, better flexibility and less harmful element contents in comparison to neat Bi₂Te₃ films. Additionally, K. Kato et al. reported both p-type and n-type flexible Bi₂Te₃ films with high ZT values of ~1, which were estimated with out-of-plane thermal conductivity.

Comment 2: *For N-type. The present study is only for p-type. N-type material is favorable to make a thermoelectric module.*

Response: A thermoelectric generator is made of both p-type and n-type materials. Thus it is necessary to explore both p-type and n-type materials with comparable thermoelectric properties. Although we only report p-type high-performance thermoelectric materials, we present a facile but robust method for preparing organic/inorganic hybrids. We are confident that n-type thermoelectric materials with high thermoelectric performance can be achieved by adopting the preparation method with n-type materials in the future work. Thanks again for your suggestions.

Comment 3: *Interfacial resistance. The electrical conductivity of the composite follow the highest predicted value in the present study, however, those values follow the lowest value mostly mentioned below (sometime, lower than the lowest value due to interfacial electrical resistance). B. Zhang et al., ACS Applied Materials & Interfaces, Vol.2, No.11 (2010) pp.3170-3178. On the other hand, interfacial thermal resistance is very high. The order is 10⁻⁵ K/W/m². Thermal resistance is well reviewed in the following article. The order of the resistance is 10⁻¹⁰*

to 10^{-7} K/W/m². T. Zhan et al., *Scientific Report*, Vol.7 (2017) 7109. The electrical resistance is really low although the thermal resistance is 100 times higher than other reported value at the present study. The mechanism of the present study should be explained to have the readers' understanding.

Response: Thank you very much for your comment.

(1) Regarding to the electrical conductivity. We also noticed the results reported by B. Zhang et al. (*ACS Applied Materials & Interfaces*, 2010, 2, 3170), and mentioned this reference (Ref. 13) in the Introduction. In Ref. 13, the inorganic particles severely aggregated and sank to the bottom of hybrids, forming a two-layer structure. This made the electrical conductivity of hybrids far below the calculated values based on mixture models. On the other hand, the electrical conductivity (1000 S cm^{-1}) of Bi₂Te₃ used in the mixture models was taken from conventional sintered Bi₂Te₃ bulk, which should be very different with their samples that were prepared by simple solution process. As to our samples, the electrical conductivity of hybrid films gradually approached to the lowest predicted values with increased Bi₂Te₃ nanoparticle fraction (Fig. 3a). It was mainly caused by the increase of the interfacial surface-to-volume ratio which scattered more carriers, and thereby decreasing the electrical conductivity.

(2) Regarding to the interfacial thermal and electrical resistance. Thank you very much for providing us with this useful article (*Scientific Report*, 2017, 7, 7109). This paper well reviewed the thermal resistance for lots of selected hybrid systems, and the order of the resistance is from 10^{-10} to $10^{-7} \text{ m}^2 \text{ K W}^{-1}$. However, there are also lots of reported thermal resistances which are in the similar order (10^{-6} to $10^{-5} \text{ m}^2 \text{ K W}^{-1}$) with our sample. (Ref: (1) *International Journal of Thermal Sciences*, 2009, 48, 2221. (2) *Journal of Power Technologies*, 2014, 94, 270. (3) *Science Bulletin*, 2018, <https://doi.org/10.1016/j.scib.2018.02.022>.) Generally, the phonon mean free path is much longer than that of charge carriers. (Ref: *Advanced Functional Materials*, 2017, 1702847.) The size of nanoparticles in the hybrids may be similar to the mean free path of phonons but much longer than that of charge carriers. Therefore, thermal transport can be significantly suppressed due to the boundary phonon scattering effect, while electrical transport is slightly affected. (Ref: *Advanced Materials*, 2007, 19, 1043.) Consequently, the electrical resistance is low although the thermal resistance is high. All the discussions are added and highlighted on Page 11 in the revised manuscript.

Reviewers' comments:

Reviewer #1 (Remarks to the Author):

The authors have addressed all of my concerns. I think that now the manuscript can be accepted for publication in Nat Comm. One thing I want to stress again that the all the thermoelectric properties for zT calculation should be measured on the same sample along the same direction, otherwise the zT value may have a large deviation from the true one. In this work, the thermal conductivity was measured on a thick film but the electrical properties were measured on a thin film. The authors should bare mind that such measurements may result in a relatively large error for zT estimation.

Reviewer #2 (Remarks to the Author):

I suggest acceptance of the manuscript since appropriate revision has been made according to the reveiwer's comments.

Reviewer #3 (Remarks to the Author):

Thank you for the response, but I have the following questions.

(1) Dr. Kato made a Bi₂Te₃-PEDOT:PSS thick film. Out-of-plane thermal conductivity was used to calculated the ZT, but their film was enough thick for thermal conductivity measurement of the composite. Sub-micron structure smaller than the thickness was made in the film. Their ZT estimation was not bad.

(2) I understand the response for comment 2.

(3) I have read the papers pointed out in the response. I found that $10^{-5} \text{ m}^2 \text{ K/W}$ order thermal resistance had been reported. In one of the papers(Journal of Power Technologies, 2014, 94, 270), the authors explained that the high thermal resistance is caused by the imperfections at the interfaces in Line 2 on Page 279. In the present study, authors should explain the low electrical interfacial resistance mechanism at the imperfection interface.

In addition, low thermal conductivity is explained by long phonon mean free path in the response. However, the reported lattice thermal conductivity of Bismuth Telluride nano-wire with 52nm diameter is about 0.5 W/m/K (Journal of Applied Physics, 2009, 105, 104318). They explained that the thermal conductivity reduction was only 20% due to the short phonon mean free path in Bi₂Te₃.

We roughly calculated lattice thermal conductivity by using Lorentz number in the present study. The calculated lattice thermal conductivity was about 0.4 W/m/K for PEDOT/Bi₂Te₃(100). The measured thermal conductivity in the present study was lower than that of nano-wire. More explanation is necessary for the publication.

Reviewer #1:

Comment: *The authors have addressed all of my concerns. I think that now the manuscript can be accepted for publication in Nat. Commun.. One thing I want to stress again that the all the thermoelectric properties for zT calculation should be measured on the same sample along the same direction, otherwise the zT value may have a large deviation from the true one. In this work, the thermal conductivity was measured on a thick film but the electrical properties were measured on a thin film. The authors should bare mind that such measurements may result in a relatively large error for zT estimation.*

Response:

Thank you for your recommendation. Our preparation method is suitable for preparing organic/inorganic hybrid films with thickness less than 100 nm, so the electrical conductivity and Seebeck coefficient were measured from thin films. We agree that the ZT evaluation should be performed on the same sample. However, to the best of our knowledge, it is still lack of accurate methods for performing the in-plane thermal conductivity measurement of nano-thick films with substrates. Thus, we prepared thick films (thickness of $\sim 0.75 \mu\text{m}$) for the 3ω thermal conductivity measurement. This also is a common solution in previous works (*Nat. Mater.*, 2011, 10, 429. *Nat. Mater.*, 2013, 12, 719. *J. Am. Chem. Soc.*, 2017, 139, 13013.). The total thermal conductivity contains the lattice thermal conductivity and electronic thermal conductivity. In our hybrid films, the lattice thermal conductivity is mainly dominated by the film morphology. The thick films for the 3ω thermal conductivity measurement display a very similar morphology as compared to the corresponding thin films. For example, the SEM image of as-prepared PEDOT/31 vol% $\text{Bi}_2\text{Te}_3(100)$ hybrid thick film is given in Supplementary Fig. 10. While the electronic thermal conductivity is related to the electrical conductivity. Our thick films for thermal conductivity measurement only possesses 10~20% lower electrical conductivity as compared to the corresponding thin films. For example, the electrical conductivity of PEDOT/31 vol% $\text{Bi}_2\text{Te}_3(100)$ hybrid thick film is $\sim 391 \text{ S cm}^{-1}$, which is $\sim 483 \text{ S cm}^{-1}$ for the corresponding thin film. It may not result in a relatively large error for the electronic thermal conductivity. Therefore, the total thermal conductivity of thick film may have limited deviation from the thin one. We have added the above discussion in the revised supplementary information (Page 10). Thanks again for your very useful suggestions.

Reviewer #2:

Comment: *I suggest acceptance of the manuscript since appropriate revision has been made according to the reviewer's comments.*

Response:

Thank you very much for your recommendation.

Reviewer #3:

Comment 1: *Dr. Kato made a Bi₂Te₃-PEDOT:PSS thick film. Out-of-plane thermal conductivity was used to calculate the ZT, but their film was enough thick for thermal conductivity measurement of the composite. Sub-micron structure smaller than the thickness was made in the film. Their ZT estimation was not bad.*

Response:

Thank you for your instructive comment. We agree with you. In Dr. Kato's paper (Journal of Electronic Materials, 2014, 43, 1733), they prepared both p-type and n-type Bi₂Te₃ films on insulating porous polyimide (PI) substrates to obtain flexible films. It is an interesting work, which provides an effective way for preparing high-performance and flexible Bi₂Te₃ films. Differently, we offer another way for making flexible organic/inorganic hybrids with high thermoelectric properties in our work. The matrix (highly conductive PEDOT) is organic component, which holds intrinsically low density and excellent flexibility.

Comment 2: *I understand the response for comment 2.*

Response:

Thank you.

Comment 3: *I have read the papers pointed out in the response. I found that 10⁻⁵ m² K/W order thermal resistance had been reported. In one of the papers (Journal of Power Technologies, 2014, 94, 270), the authors explained that the high thermal resistance is caused by the imperfections at the interfaces in Line 2 on Page 279. In the present study, authors should explain the low electrical interfacial resistance mechanism at the imperfection interface.*

In addition, low thermal conductivity is explained by long phonon mean free path in the response. However, the reported lattice thermal conductivity of Bismuth Telluride nano-wire with 52nm diameter is about 0.5 W/m/K (Journal of Applied Physics, 2009, 105, 104318). They explained that the thermal conductivity reduction was only 20% due to the short phonon mean free path in Bi₂Te₃. We roughly calculated lattice thermal conductivity by using Lorentz number in the present study. The calculated lattice thermal conductivity was about 0.4 W/m/K for PEDOT/Bi₂Te₃(100). The measured thermal conductivity in the present study was lower than that of nano-wire. More explanation is necessary for the publication.

Response:

Thank you for your instructive comments. We have made further explanation for the high interfacial thermal resistance and low interfacial electrical resistance at the organic/inorganic hybrid interface in the revised manuscript.

The low interfacial thermal conductance is possibly caused by the strong acoustic mismatch due to different phonon densities and velocities between polymer matrix (PEDOT) and inorganic filler (Bi₂Te₃) (Ref. 3,43 in the revised manuscript). Furthermore, the phonon mean free path in PEDOT might be in the order of 10¹~10² nm as simulated in Ref44, which is comparable to the size of Bi₂Te₃ nanoparticles in the hybrid films. This will enhance interfacial phonon scattering and thereby the thermal transport can be considerably suppressed (Ref. 3,12).

Regarding to the electrical transport, the highly conductive PEDOT matrix in hybrids is continuous, providing hole transport paths. More importantly, PEDOT is *in situ* polymerized with the presence of Bi₂Te₃ nanoparticles. This is helpful for intimate contact between PEDOT and Bi₂Te₃ and thereby enhancing charge transport across the PEDOT-Bi₂Te₃ interfaces (Ref. 33). So, we do not think the PEDOT-Bi₂Te₃ interfaces are imperfection. Many works have reported *in situ* synthesized polymer composites with enhanced electrical transport properties (Ref. 3,45). Besides, the mean free path of holes in PEDOT was calculated to be 10⁰~10¹ nm (Ref. 44), which is much smaller than the size of Bi₂Te₃ nanoparticles (~40-500 nm), suggesting a negligible degradation to carrier transport. Thus, the hole transport can be minimally affected as compared to the phonon transport.

Experimental measurement of phonon mean free path, such as using frequency domain thermoreflectance method, is helpful to illustrate the interfacial thermal/electrical conductance in the future work. (Nature Communications volume 4, 1640 (2013))

The paper (Journal of Applied Physics, 2009, 105, 104318) reported the thermal conductivity of *individual bismuth telluride nanowires* with a 52nm diameter, where thermal transport is along the continuous single nanowire. That is significantly different from our research here. Our Bi₂Te₃/PEDOT hybrid film consists of numerous **discontinuous and isolated** Bi₂Te₃ particles embedded in the PEDOT polymer matrix. The thermal transport mechanism should be very different. The factors on the thermal conductivity of our hybrid film are complex, including the volume fraction, morphology, the thermal conductivity of conducting polymer and the inorganic fillers, and especially the interfacial thermal conductance. The interfacial effect may contribute to the lower lattice thermal conductivity of our hybrid sample as explained above.

The modifications are highlighted on Page 11 in the revised manuscript. We hope our revision can answer your comments. Thank you very much again for your valuable comments.

REVIEWERS' COMMENTS:

Reviewer #3 (Remarks to the Author):

The authors have addressed all of my concerns. I think that now the manuscript can be accepted for publication in Nat. Commun..

Reviewer #3:

Comment: *The authors have addressed all of my concerns. I think that now the manuscript can be accepted for publication in Nat. Commun..*

Response:

Thank you very much for your recommendation.